# The Use of Partial Least Squares–Path Modelling to Understand the Impact of Ambivalent Sexism on Violence-Justification among Adolescents

**DOI:** 10.3390/ijerph17144991

**Published:** 2020-07-10

**Authors:** Roberto Fasanelli, Ida Galli, Maria Gabriella Grassia, Marina Marino, Rosanna Cataldo, Carlo Natale Lauro, Chiara Castiello, Filomena Grassia, Caterina Arcidiacono, Fortuna Procentese

**Affiliations:** 1Department of Social Science, University of Naples “Federico II”, 80138 Naples, Italy; idagalli@unina.it (I.G.); mariagabriella.grassia@unina.it (M.G.G.); marina.marino@unina.it (M.M.); rosanna.cataldo2@unina.it (R.C.); 2Department of Economics and Statistics, University of Naples “Federico II”, 80126 Naples, Italy; natale.lauro@unina.it; 3Institute for Socio-Psychological Training, Research and Sexual Education, 80127 Naples, Italy; c.chiaracastiello@gmail.com; 4Istituto Nazionale di Statistica, 00184 Rome, Italy; grassia@istat.it; 5Department of Humanities, University of Naples “Federico II”, 80138 Naples, Italy; caterina.arcidiacono@unina.it (C.A.); fortuna.procentese@unina.it (F.P.)

**Keywords:** adolescents, sexism, violence legitimation, Partial Least Squares–Path Modelling

## Abstract

Gender violence is generally conceived as a phenomenon concerning only adults. Nonetheless, it is also perpetrated within teenagers’ relationships, as many empirical studies have shown. We therefore have focused our attention on a non-probabilistic sample consisting of 400 adolescents living in Naples (Italy), to study the association between sexism and the justification of violent attitudes. Generally, sexism is recognised as a discriminatory attitude towards people, based on their biological sex. However, it is conventional to talk about sexism as a prejudice against women. The Ambivalent Sexism Inventory (ASI) for adolescents was used to evaluate the two dimensions of ambivalent sexism, i.e., hostile sexism (HS) and benevolent sexism (BS). Moreover, the questionnaire regarding attitudes towards diversity and violence (CADV) was administered to assess participants’ attitudes towards violence. A Partial Least Square–Second Order Path Model reveals that girls’ ambivalent sexism is affected more by benevolent sexism than hostile sexism. On the contrary, among boys, hostile sexism has a higher impact. Finally, benevolent sexist girls justify domestic violence more than boys do.

## 1. Introduction

Is sexism a prejudice? Glick and Fiske [1] (p. 491) wrote: “Sexism is indeed a prejudice, but a special case of prejudice marked by a deep ambivalence, rather than a uniform antipathy, toward women. Sexism has typically been conceptualized as a reflection of hostility toward women. This view neglects a significant aspect of sexism: the subjectively positive feelings toward women that often go hand in hand with sexist antipathy”. Glick and Fiske [1] defined sexism as a multidimensional construct, labelled “ambivalent sexism”, which includes both a negative perception of women-hostile sexism, and a positive one, termed benevolent sexism. This recognises that sexism entails a mixture of hostile and subjective benevolence, comprised of these two distinct yet complementary ideologies.

Hostile sexism is reflected by misogyny (a hatred of women), and is expressed through blatant negative evaluations of women. It may include the beliefs that women are incompetent, unintelligent, overly emotional and sexually manipulative. On the other side, hostile sexism views gender relations in combative terms; women are seen as seeking to usurp men’s power in various ways, via their sexuality, by claiming discrimination, or through feminist activism.

Benevolent sexism is reflected in seemingly positive evaluations of women. It may include the representation of women in roles such as wife, mother and child caregiver, the romanticising of women, and the belief that men must protect women, but it also depicts women as weak and in need of male protection, thus emphasising women’s lower status [2,3]. However, the conception of ambivalent sexism proposed by Glick and Fiske [1] implies that hostile and benevolent sexism may be positively correlated.

According to Connor et al. [2], hostile and benevolent sexism are not conflicting, but are complementary ideologies that present a resolution to the gender relationship paradox. Benevolent sexism recruits women as unconscious participants in their own submission, thereby limiting apparent coercion by offering male protection to women in return for their acquiescence. Hostile sexism preserves the existing conditions by penalising those who don’t respect traditional gender roles.

Glick and Fiske [1] (p. 492) wrote: “If the two sets of beliefs are positively correlated, can they be called “ambivalent”? We characterize them as ambivalent because, even if the beliefs about women that generate hostile and benevolent sexism are positively related, they have to oppose evaluative implications, fulfilling the literal meaning of ambivalence (both valences)”. They [1] described ambivalent sexism for the adult population and provided a validated measure of ambivalent sexism. Then, Lemus et al. [4] adapted this measure for adolescents.

Hostile and benevolent sexism each include subcomponents related to paternalism (the notion that men should rule), gender differentiation (distinguishing the genders through roles and stereotypes) and heterosexual intimacy (sexuality and intimate relationships). The paternalistic component goes directly to the heart of the issue, supporting men’s power and status. Hostile sexism’s dominative paternalism supports men’s greater power. In contemporary societies, women’s experienced elevations of status entail men seeing women as threatening, and trying to steal men’s power and gain unfair advantages over men. Benevolent sexism’s protective paternalism component justifies limiting women’s access to resources and masculine roles as necessary in order to protect and to serve their best interests [2]

Sexism is linked to other prejudices, such as racism, xenophobia, religious intolerance and heterosexism. They are all connected because, as Blummenfeld and Raymond [5] say, they “all involved a negative prejudgment whose purpose is to maintain control and power”. Pharr [6] defines these social phenomena as each having the ability to control and destroy lives. Racism, xenophobia and sexism are also combined in the experiences of women of colour and immigrant women. Collins [7], for example, argues that African American women in the United States live in a state of triple oppression, by race, sex and class, with these oppressions being articulated both by the white community and within the black community. The persistence of discrimination related to gender issues is a consistent problem in contemporary society [8,9,10]. Rollero et al. [10] (p. 4) highlight that, in “the perspective of social dominance theory, both Hostile Sexism and Benevolent Sexism may be conceptualized as myths that support gender hierarchy”. Consistent with this, empirical research has found that Social Dominance Orientation predicts both hostile sexism and benevolent sexism [11,12], as hostile sexism promotes male dominance over women and benevolent sexism justifies women’s subordinate status [13].

Originally, it was believed that the phenomenon of gender violence only concerned adults. Nowadays, numerous empirical studies have shown that it is also perpetrated within relationships described as relating to teenagers.

Recent systematic reviews highlight the topic, showing that adolescents who have more sexist attitudes exhibit a greater acceptance of intimate partner violence, greater sexual risk behaviours, greater attraction to sexist partners, greater support for the myth of idealised love and the myth of the love–abuse link, greater emotional dependence on the partner, and a poorer quality of relationships. Moreover, these studies have revealed gender-based differences in some of the mentioned variables [14].

Sexism and violence do not only concern adults, but also, and above all, adolescents. Indeed, adolescence is the moment we begin to structure contact with the “other” in a more meaningful way, on a relational and sexual level. More attention should be focused on how this contact is perceived and actualised. It is therefore the moment when every action regarding the awareness and prevention of gender-based violence should be focused on. Violent behaviour in adolescents is a crucial social problem due to its serious consequences.

Carrascosa et al. [15] analysed the differences between boys and girls in their violent behaviour with respect to their peers. In their study, boys revealed greater acceptance of both transgressions of rules, and direct as well as indirect violence. On the other hand, girls reached the highest scores concerning open communication with their mothers.

Other research on adolescent behaviour has shown that prejudices affect perceptions of violence and shape its definition [16,17], while some stereotypical beliefs about men’s and women’s roles, diversity, and minorities, may justify violent attitudes [18,19,20].

In Spain, some authors specifically adapted a new research tool to the young population [21]. In fact, there, as well as in several Latin American countries, the problem of sexism among young people has attracted a great deal of attention. In particular, Lopez, Chesney-Lind and Foley [22] indicate that boys/men used several strategies to control girls, and girls used a variety of strategies to resist their control actions. Marta Ferragut and colleagues [23] investigated sexism among boys and girls, and found that boys are more inclined to justify attitudes of violence, showing greater agreement with sexist beliefs. Moya Morales [24] argued that ambivalent sexism may pave the way toward domestic violence. Recently, in the Anglo-American literature, dating violence via the use of digital technologies and social media has received increased attention [25,26,27,28,29]. In Italy, Procentese [30] investigated this issue, focusing on partners’ behaviour in young couples and examining the acceptance of traditional gender roles. The well-known “gender ideology” (patriarchy and masculinism) [24], which promotes asymmetrical relationships of power between men and women, is often the basis or the driver of violent behaviours. The power asymmetry, onto which gender asymmetry is grafted, represents a critical variable in predicting couple violence. In Procentese’s study, couples with a violent male partner were more frequently asymmetrical as far as power was concerned. Other interesting Italian studies relate to sexting, and dating violence and its justification [31,32]. Nevertheless, further research, specifically on sexism among young people/couples, is needed.

Starting from these premises, the present contribution has the following main aims:Aim 1—Analysing the relationship between sexist attitudes (hostile sexism, benevolent sexism and ambivalent sexism) and attitudes towards diversity and violence;Aim 2—Analysing the sexist attitudes of Naples’s adolescents to find gender differences.

The below hypotheses summarise the core proposals.

**Hypothesis** **1**—*Hostile sexism and benevolent sexism are structured differently in adolescent boys and girls. Therefore, ambivalent sexism has a different composition (i.e., the hostile dimension can weigh more than the benevolent one on attitudinal ambivalence, and vice versa) depending on gender*.

**Hypothesis** **2**—*Hostile sexism, benevolent sexism and ambivalent sexism have a strong impact on the justification of peer violence, domestic violence and violence against minorities. A different impact for boys and girls is expected*.

Díaz-Aguado et al. [33] used a questionnaire to assess different kinds of beliefs that could lead adolescents to justify exclusion and aggression. This questionnaire is called the questionnaire of attitudes toward diversity and violence (in Spanish “Cuestionario de Actitudes hacia la Diversidad y la Violencia”; CADV).

In this study, using the aforementioned Inventory of Ambivalent Sexism [4] and the questionnaire regarding attitudes towards diversity and violence [33] on a sample of adolescents living in Naples, we have formalised the relationship between sexist and violent attitudes through a Structural Equation Model [34,35]: “hostile sexism” and “benevolent sexism”, with the dependent variables “justification of peer violence,” “justification of domestic violence” and “justification of violence against minorities”, are latent variables that we have inferred from the available ones. All these notions are latent concepts that are not directly measurable. They identify complex and multidimensional phenomena, and their relative descriptions are available through a suitable synthesis of the associated manifest variables that are their elementary indicators. In this case, the latent variables are dimensions of the questionnaire, while the manifest variables are their relative items (see also Section 2.1 for scales reliability).

## 2. Materials and Methods

The study presented here derives from a pilot study conducted by a research group from the Department of Social Science of the University of Naples “Federico II”, in the second semester of the 2017–2018 school year [32,36,37]. It investigated some opinions, beliefs, attitudes and representations about teenage behaviours, in general, and their intimate relationships, in particular, involving Neapolitan students.

For the research presented here, a questionnaire was administered to 400 teenagers attending the last two years of eight public-funded high schools in Naples, in order to investigate their sexist attitudes as well as their possible tendency to justify gender-based violence (see Section 2.1 for measure instruments). Of the sample, 56% were women, 76% were aged 17 or 18 (*Md* = 17.73; *St.Dev.* = 0.925) and 62% were attending the last year. Of the sample, 30% had two working parents, while 40% had a mother who did not work outside the home.

Before giving the questionnaire to the students, we met with their teachers and parents who provided the necessary consent for the students’ participation in the study, in accordance with the Italian Psychological Association ethical guidelines [38]. In the case of minors, the code of ethics of the Italian Association of Psychology requires that both parents express their consent. We respected this rule. In addition, we clearly informed the participants that consent could be given, refused or withdrawn at any time. They were granted the freedom to make decisions and had all the time necessary to reflect, raise doubts and ask for clarification. The interviewer explicitly informed those who participated in the research that they were free to skip questions or withdraw at any time without having to give any justification, and that refusal to participate or the decision to withdraw would not be prejudicial.

Students were administered questionnaires in the classroom collectively and each student completed the questionnaire anonymously in a dedicated time during class hours.

The questionnaires with more than 50% of the data missing were eliminated (40 questionnaires); finally, 360 questionnaires were collected and used for the analysis. The “nearest neighbour” (which consists of introducing a concept of similarity between the units, based on a distance function) algorithm was used to estimate missing data.

### 2.1. Measurement Instruments

The questionnaires were self-administered and students were contacted at the school between classes. The procedure generally takes between 40 and 60 min. There were no incentives offered to participants [39].

The semi-structured questionnaire included scalar items, checklists and open-ended questions. It was organised in six blocks. In the first block respondents provided information about their sex, age, school, class, father’s and mother’s occupations, and religious and political orientations.

The next two blocks came from the Ambivalent Sexism Inventory (ASI) for adolescents [4]. The items, which were rated on a 6-point Likert-type response format, range from 1 (“strongly disagree”) to 6 (“strongly agree”). The themes were:Hostile sexism (10 items), reflecting negative attitudes towards women;Benevolent sexism (10 items), reflecting protective attitudes towards women, but attitudes which are stereotyped and depreciative.

The scales’ reliability measures obtained in the current study are more robust (HS *α* = 0.84; BS *α* = 0.79; AS *α* = 0.87) than the ones (HS *α* = 0.81; BS *α* = 0.84; AS *α* = 0.77) from the original study [4].

The last three blocks came from the questionnaire on attitudes towards diversity and violence [33]. The items, which were rated on a 7-point Likert-type response format, range from 1 (“strongly disagree”) to 7 (“strongly agree”).

The themes were:F1-Sexist beliefs and the justification of domestic violence (17 items), justifying the man as the head of the family, sexist discrimination, child abuse, violence toward women and intolerance (called the “justification of domestic violence”);F2-Justification of peer violence as reactive and courageous (18 items), associating peer violence with displays of courage (called the “justification of peer violence”);F3-Intolerance and the justification of violence against minorities as a punishment (14 items), xenophobia, racism, the rejection of tolerance and diversity, and the justification of violence against minorities perceived as different (called the “justification of violence against minorities”).

In terms of psychometric properties, the factors related to the justification of violence showed a high internal consistency (CADV_F1 *α* = 0.89; CADV_F2 *α* = 0.86; CADV_F3 *α* = 0.84). These results are also stronger than the ones provided by the original validation study (CADV_F1 *α* = 0.85; CADV_F2 *α* = 0.85; CADV_F3 *α* = 0.82) [33].

Both the ASI scale and the CADV scale used in this study were the Italian version [32].

### 2.2. Methodology

We considered the blocks with scalar items (hostile sexism; benevolent sexism; the justification of peer violence; the justification of domestic violence; and the justification of violence against minorities) as additive scales. For each unit *i*, the score of the *q*-th additive scale was:(1)$${\mathrm{scale}}_{iq}=\overset{P}{\underset{p=1}{\sum }}\;x_{ip},$$
where P = number of items of the *q*-th scale (1).

We verified that, for each block, all items measured the same concept, through Cronbach alpha [40].

First, the two scales were normalised with scores from 0 to 100 to make them homogeneous. Since the questionnaire was composed of CADV items in a Likert Scale with a range from 1 to 7, and ASI items in a Likert Scale with a range from 1 to 6, the normalisation made the data belonging to the different variables comparable. So, we transformed the data for each unit *i* and each scale *q* to compare the means of the different additive scales:(2)$$scale_{iq}^{100}=\frac{scale_{iq}-min\;(scale_{q})}{max\;(scale_{q})-min\;(scale_{q})}*100,\;0\leq scale_{iq}^{100},\leq 100.$$Each scale ranges from 0 to 100. We performed a Student’s *t*-test to compare the means between boys and girls.

In this paper, our objective was to explore existing relations between every dimension of sexism and specific facets of violence-justification. In particular, the aim was to understand if and how much an increase or decrease in the level of sexism can lead to an increase or decrease in the justification for violence. Precisely for this reason, it was decided to use Structural Equation Modelling–Partial Least Squares (SEM-PLS), or Partial Least Squares–Path Modelling (PLS-PM), since unlike Structural Equation Modeling–Covariance Based (SEM-CB), it can be applied from a predictive point of view; PLS-PM is a framework for analysing multiple relationships between a set of blocks of variables (or data tables), taking into account previous knowledge (theory) of the phenomenon under analysis, to make decisions and predictions. The SEM-CB approach is primarily used to confirm (or reject) theories (i.e., a set of systematic relationships between multiple variables that can be tested empirically). In contrast, via the PLS-PM approach, the Latent Variables (LVs) estimation plays a main role. As a matter of fact, the aim of this method is to provide an estimate of the LVs in such a way that they are most strongly correlated with one another (according to the path diagram structure) and most representative of each corresponding block of Manifest Variables (MVs). While covariance-based estimators minimise the discrepancy between the empirical and model-implied variance–covariance matrix of the observable indicators to obtain the model parameter estimates, variance-based estimators, such as the PLS estimator, create linear combinations of the indicators as standing for the theoretical concepts, and subsequently estimate the model parameters. Moreover, the decision to use the PLS-PM as the methodological framework was made for other reasons. It provides the following opportunities: to estimate the hypothesised relationships without making assumptions about data distribution; to obtain, simultaneously and coherently with the estimation method, a ranking of individuals for specific indicators; to define an optimal system of weightings; to work with a large number of variables and a few observations; to estimate complex models without any problems of identification of the model; and to work with missing data, and in the presence of multicollinearity.

In this study we chose this method to derive a measure of ambivalent sexism considering its multidimensional nature. Specifically, we used the PLS-PM [41,42,43] to estimate the SEM, as shown in Figure 1. The model that arose from the hypothesis was that adolescents with sexist attitudes were more likely to justify many different types of violence. The LVs are: hostile sexism *ξ*_1_, benevolent sexism *ξ*_2_, the justification of domestic violence *ξ*_3_, the justification of peer violence *ξ*_4_, and the justification of violence against minorities *ξ*_5_. Hostile sexism and benevolent sexism are explanatory LVs; the justification of peer violence, the justification of domestic violence, and the justification of violence against minorities are dependent LVs. There is a sixth LV in Figure 1—ambivalent sexism as the union of hostile sexism and benevolent sexism. Ambivalent sexism was considered in this study as a latent and multidimensional concept, not directly measurable, linked to two constructs (hostile and benevolent sexism) representing the different dimensions of sexism. Ambivalent sexism was considered as a second-order LV [44,45], constituted by the first-order LVs of hostile and benevolent sexism.

The MVs of hostile and benevolent sexism were the items on the Ambivalent Sexism Inventory for adolescents [4], while the MVs of the justification of peer violence, the justification of domestic violence and the justification of violence against minorities were the items on the questionnaire regarding attitudes towards diversity and violence [33].

Ambivalent sexism had no manifest variables associated. Table A1 (see Appendix A) shows the MVs for each LV.

To estimate the model, we used the mixed two-step approach [46], which [47] performs better than the procedure with repeated indicators [48,49,50,51] and the two-step approach [52]. The mixed two-step approach has two steps. In the first step, this approach, like the repeated indicators approach, estimates the model via the PLS algorithm, which is then repeated for the second-order LVs, and the same MVs of the first order LVs. In the second step, it re-estimates the model using the PLS algorithm, and the MVs of the second-order LVs are the scores obtained in the first step.

We estimated the model of ambivalent sexism at the first step, repeating for ambivalent sexism the same MVs of hostile and benevolent sexism (Figure 2), and giving, at the second step, as MVs of ambivalent sexism, the scores of hostile and benevolent sexism that were obtained in the first step (Figure 3).

The scores are the result of the model estimated in the first step, considering all the manifest variables. These variables, in the second step, represent the ones associated with the ambivalent block.

The measurement model is formative [43]; each LV is generated by its own MVs, as can be seen in Figure 2 and Figure 3, where the arrows extend from the MVs to their own LV. The structural model is formative too; hostile and benevolent sexism determine ambivalent sexism and its impact on the justification of violence (Figure 1).

We used Xlstat-Plspm (Addinsoft, Paris, France) [53] with a path weighting scheme for the structural model [43]. Only for the parameter estimation phase are the MVs standardised (*µ* = 0 and *σ* = 1). We validated the results through the R square [54,55], which we considered as being “substantial” if it was greater than 0.67, “moderate” if it was between 0.33 and 0.67 and “weak” if it was between 0.19 and 0.33 [56].

We estimated the parameters of the model with a bootstrap procedure [56,57,58,59,60]. We used a procedure called “decision matrix analysis” developed by Hock et al. [61] and Schloderer et al., [62] which takes into account the average of each estimated explanatory LV and the path coefficient on each estimated dependent variable. The construction of the decision matrix analysis is a key characteristic of the PLS-PM method. This matrix is a simple and valid tool for the diagnosis and detection of such levers. It consists of a dispersion graph, which allows each variable to be positioned based on the average score (coordinated on the *y*-axis) and on the estimated impact on the target LV (coordinated on the *x*-axis). The matrix is divided into four areas: the first area is the most critical because the variables have a high impact but a low mean value; the second is the area of the monitoring, in which the variables have a low value both for the mean and the path coefficient; the third is the area to be improved because the variables have a high mean value and a low path coefficient; finally, the fourth is the area to be maintained, in which variables have a high value both for the mean and for the path coefficient.

## 3. Results

### 3.1. Internal Consistency and Descriptive Analysis of the Blocks

Table A2 shows the Cronbach’s alpha coefficients for each block composed by the scalar items (the themes of the survey).

The internal consistency (Cronbach’s alpha standardised item) ranges between the values of *α* = 0.81 and *α* = 0.90 for the total sample (*N* =360); it ranges between the values of *α* = 0.76 and *α* = 0.90 for the boys and between the values of *α* = 0.81 and *α* = 0.90 for the girls. The results show that all blocks have good internal consistency.

Table A3 shows the descriptive statistics for each of the six blocks, separately, for the boys and girls. The means of the hostile sexism scale, the benevolent sexism scale and the ambivalent sexism scale (hostile sexism = 54.8; benevolent sexism = 67.3; and ambivalent sexism = 61.1) are higher than the means of the justification of peer violence scale, the justification of domestic violence scale, and the intolerance and the justification of violence against minorities scale.

Among the six scales, boys scored higher on hostile sexism, ambivalent sexism, the justification of domestic violence and the justification of peer violence.

Table 1 shows the results of the *t*-test with equal variance not assumed. Only for the benevolent sexism scale and the justification of violence against minorities scale are there no significant differences between the girls and boys.

### 3.2. The Second-Order Structural Equation Model

We validated the results of the second-order model through the R square. Table 2 shows how all the R squares can be described as substantial for both the boys and the girls.

The results of the inner estimation are graphically presented in Figure 4, where the path coefficients of each block are reported, and Table 3 shows how all the path coefficients have a positive sign, and are significant. The hostile sexism results are more important for determining ambivalent sexism for boys than girls; for girls, hostile and benevolent sexism have a similar impact on the ambivalent construct of sexism. Ambivalent sexism has a high impact on all dimensions of justification of violence.

To verify if the impact of ambivalent sexism on the justification of peer violence, the justification of domestic violence and the justification of violence against minorities is different for boys and girls, we compared the path coefficients between the two groups.

Table 4 shows the results of the *t*-test on the path coefficients. The differences are significant for the impact of hostile sexism and benevolent sexism on ambivalent sexism, and for ambivalent sexism on the justification of domestic violence. For the girls, benevolent sexism has the same impact (0.52) as hostile sexism (0.57) on ambivalent sexism, while ambivalent sexism has a higher impact on the justification of domestic violence (0.83); for the boys, hostile sexism has a higher impact (0.85) on ambivalent sexism. This means that, for the girls, benevolent sexism, having a high impact on ambivalent sexism, represents a critical variable.

Table A4 in particular, shows the MVs that are significant for the boys and girls. For hostile sexism, the significant MVs for the boys are all the MVs (X7) except one, while for the girls, the first four (X1, X2, X3, X4) are significant. This means that both groups recognise the traditional figures of boys and girls and their relative roles; for example, boys are physically stronger than girls and they should exert control over who their girlfriends interact with. On the other side, girls should help their mothers at home more than boys, and they are better at domestic tasks, whereas boys are more skilled at fixing things. If we look at benevolent sexism, a boy has to protect a girl and must be romantic with her (the significant MVs for the boys are X11 and X19, while for girls they are X15, X17 and X20). For the justification of domestic violence, the strong and powerful figure of the boy is recognised on both sides: it is the boy who makes decisions in the family, who must work and who must not cry (the significant MVs for the boys are X23, X24, X25, X26 and X37, while for girls they are X27, X33, X34 and X37). For the justification of peer violence, boys recognise violence as part of human nature, and are willing to react to protect their “property” (the significant MVs for the boys are X38, X41, X49, X50 and X53, while for girls they are X39, X43, X46 and X50). For the justification of violence against minorities, both groups do not have much tolerance towards immigrants or people of other cultures and religions (the significant MVs for the boys are X55, X57, X61, X65, X66 and X67, while for girls they are X55, X56 and X65), and everyone is convinced that a group that tolerates too many differences of opinion cannot last long (X65).

Figure 5 shows the decision matrix of benevolent sexism, hostile sexism and ambivalent sexism with relation to the justification of domestic violence, as Table 4 shows that the difference between boys and girls is particularly significant in this block.

The decision matrix analysis shows that all constructs have a high mean and high impact, except benevolent sexism for boys, which has high a mean but low impact; this block is thus part of the area to improve. This analysis confirms what has been said in Table 4.

## 4. Discussion and Conclusions

The present study aimed at exploring the association between sexist attitudes and the justification of violence among adolescents. In particular, it was hypothesised that ambivalent sexism has a different configuration for boys and girls. Moreover, all the dimensions of sexism have a strong impact on different forms of violence-justification (peer, domestic and against minorities). Furthermore, a different impact for boys and girls was expected.

First of all, this investigation, focused on adolescents attending the last two years of eight public-funded high schools in Naples, seems to confirm that sexism and the justification of violence do not only apply to adults. Indeed, they are relevant aspects within adolescent relationships, as shown by Díaz-Aguado [18], González-Ortega et al. [63], Giordano et al. [64], Hernández [65], Merino et al. [66] and Rollero et al. [10].

All the scales present significant differences between girls and boys. Only for the benevolent sexism scale and the justification of violence against minorities scale are there no significant differences between the two groups.

In general, hostile sexism reveals itself to be more important for boys in determining ambivalent sexism than for girls. Furthermore, as regards girls, hostile and benevolent sexism have a similar impact on the ambivalent construct of sexism. Ambivalent sexism has a high impact on all dimensions of the justification of violence.

Boys and girls, however, express their sexism differently, i.e., girls’ ambivalent sexism is affected more by benevolent sexism than hostile sexism. On the contrary, among boys, hostile sexism has a higher impact, and there is a major disparity in their view of the two types of sexism. Last but not least, benevolent sexist girls justify domestic violence more than boys do.

With regard to the results concerning boys, sexist beliefs, perpetuating traditional roles and controlling behaviours belong to the analytical category of “micromachism” (micro-chauvinism) [67]. This is a concept that underpins all those subtle daily actions that constitute a real strategy of control and micro-violence aimed at undermine women’s autonomy. These manifestations of violence are often invisible, or worse, fully legitimised by the social environment.

These results are consistent with previous studies [68,69], which, using the same instruments and similar samples, found significant positive correlations among all kinds of sexist attitudes and the justification for all forms of violence. Nevertheless, in our case the most alarming result involves girls. Female participants, indeed, agree with beliefs and sexist attitudes, as well as the justification for violence against women. In other words, surprisingly enough, they do not recognise these manifestations of violence as dangerous, thereby accepting and even facilitating them. The adherence to sexist ideas and beliefs, together with an inclination towards forgiveness and justification, which has been found among the female adolescents in this study, could be linked to the tendency of women victims of violence to minimise the responsibility of the aggressor. Those are, actually, the same women who endure a violent relationship, do not report the violent “lover”, and always hope for a positive change in the violent partner [32].

To summarise, sexism (either hostile or benevolent) has an impact on the justification of violence. This impact is different according to gender. Additionally, the dimensions affecting the justification for violence are different for boys and girls. Thus, training actions, educational policies or social advertising, intended to fight the gender gap and prevent violent behaviours, must be designed differently for each of the sexes.

For example, the manifest variable X13 of the benevolent sexism block (*Boys should take care of girls*) has a high weight for girls, while it is negative for boys (Table A4). Consequently, in order to carry out an awareness campaign that aims to be effective, it is necessary to start from the meanings, beliefs and representations concerning the concept of care that girls learn from the dominant culture in their life context. This is due to the fact that taking care of someone, in certain social environments, could assume the structure of domination and even of territory control, as is shown in studies carried out with a mixed-methods approach to these same issues [32].

Taking into account the above-described awareness-raising interventions, to successfully obtain a decrease in the values of some indicators, such as the manifest variable X13, would lead to a reduction of benevolent sexism. Consequently, a decrease in ambivalent sexism, the impact of which is very strong (0.83) on the justification of domestic violence, will be observed.

For this reason, it is of the utmost importance to design social advertising and prevention interventions following the specific distinctive male and female characteristics of sexism and justifications of violence. Only by dismantling the foundations on which these different social groups build the social anchoring for their attitudes, prejudices and behaviours will it be possible to achieve actual results in fighting against couple/dating violence.

### Limitations and Future Developments

Analysis of subjects from a limited geographic region reduced generalisability. The subjects were teenagers attending eight public-funded high schools in the city of Naples (southern Italy). The analytic model proposed here could be applied across a wider geographic area, in order to understand the phenomenon of sexism throughout Italy, and to further identify regional and contextual influences on sexism in adolescents. For example, north–south, city–province and centre–periphery comparisons might identify differences related to cultural and educational gradients. Southern women, according to a recent report by the Association for the Development of Industry in southern Italy [70], experience lower levels of “positive” work flexibility, they have lower job stability, and there is a lower prevalence of university graduates among women in the south than the central north of the country. Future work could also address whether residential areas [71] influence the relationships between sexist attitudes [72] and violence-justification, as foreshadowed by existing data from the National Statistical Report on violence against women [73]. Lastly, there is a need to introduce different kinds of indicators in our models. The authors’ intention is to extend the analysis to a wider population and, above all, to consider not only subjective perceptions and self-reports of adolescent sexism, but also objective environmental data on the indicators of violence and other analytical dimensions. Projects along these lines are underway.

Our results also suggest some implications for prevention interventions, which should aim at changing attitudes toward gender roles in adolescents. Consequently, interventions should not only target the males’ sexist attitudes, but also those of girls who believe they must be submissive to boys; they should also develop an ecological intervention [74,75]. A further aim should be the improvement of school and health personnel competence in dealing with sexism, with particular focus on adolescents, since there is a strong need for young people to enhance their reflexivity and positionality awareness [76,77,78,79].

## Figures and Tables

**Figure 1 ijerph-17-04991-f001:**
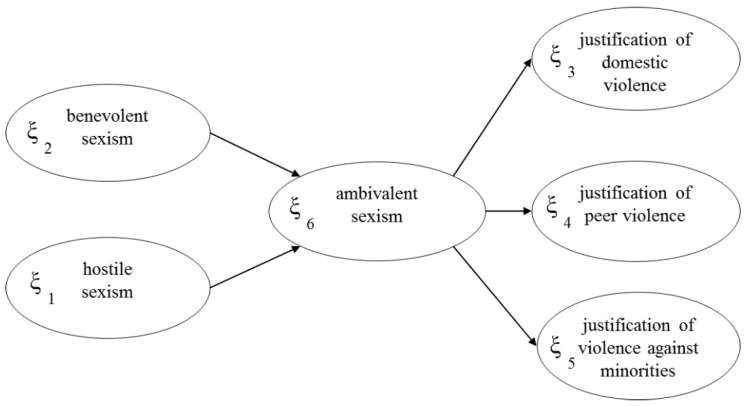
The structural model of ambivalent sexism: relationships between latent variables.

**Figure 2 ijerph-17-04991-f002:**
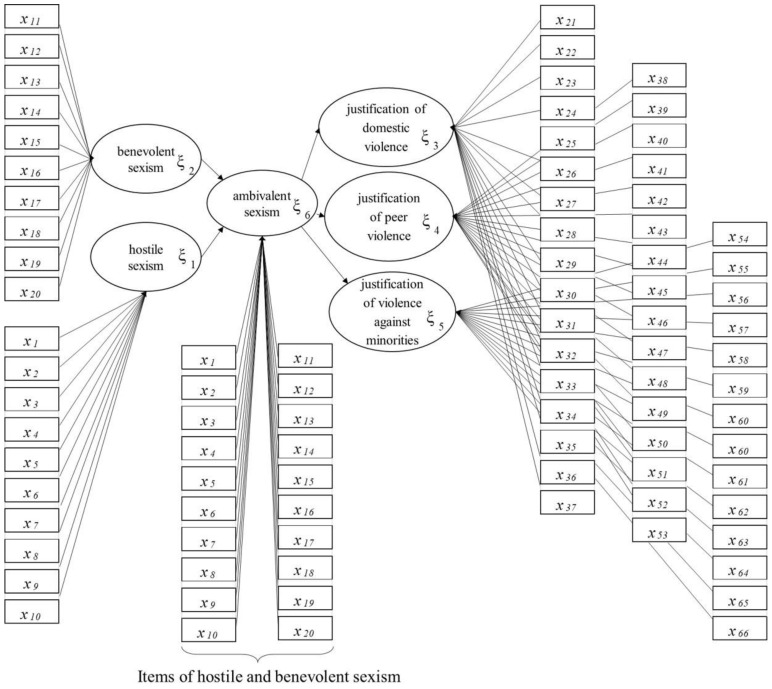
Structural and measurement model: the first step of the mixed two-step approach with the repeated MVs for the ambivalent sexism LVs.

**Figure 3 ijerph-17-04991-f003:**
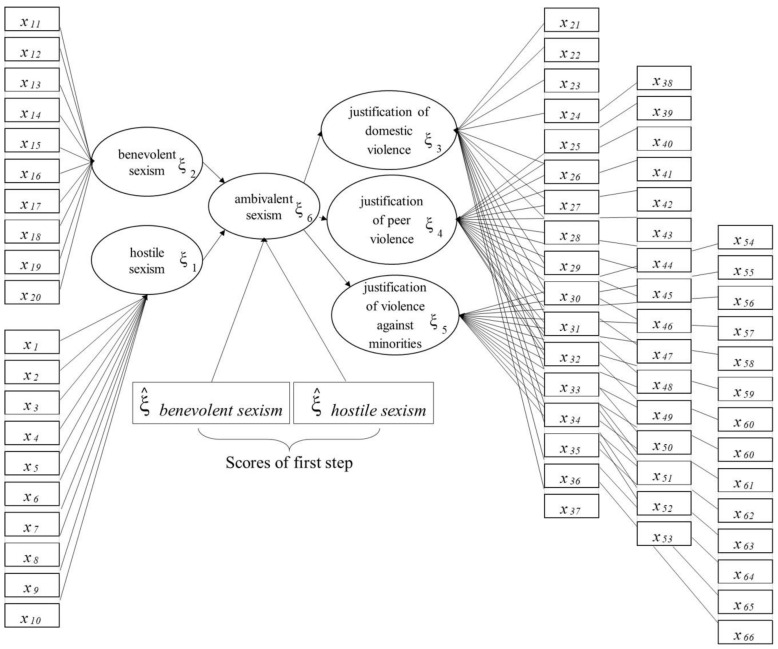
Structural and measurement model: the second step of the mixed two-step approach with the scores of first step as the MVs for the ambivalent sexism LVs.

**Figure 4 ijerph-17-04991-f004:**
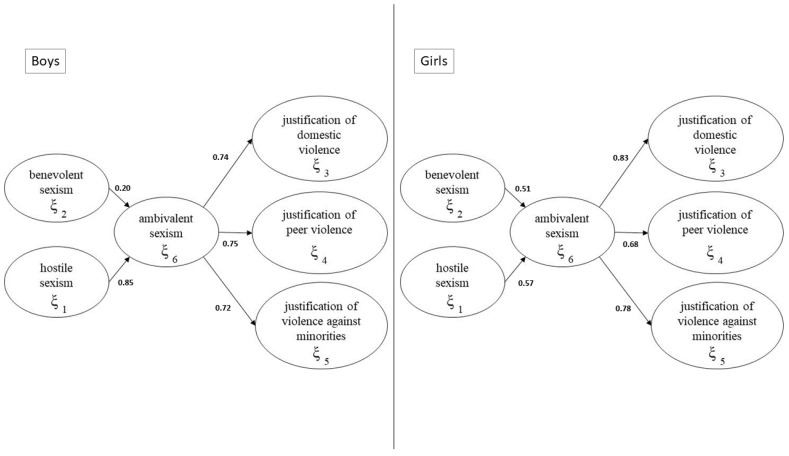
The estimated model.

**Figure 5 ijerph-17-04991-f005:**
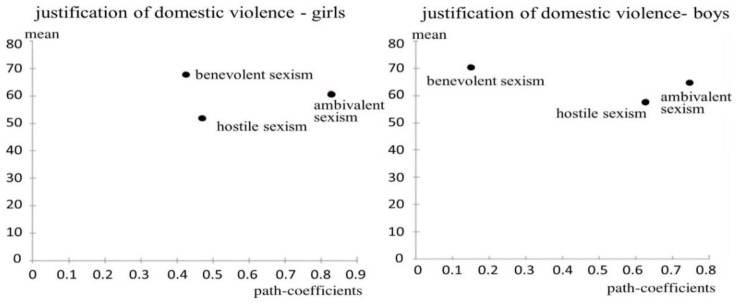
The decision matrix analysis.

**Table 1 ijerph-17-04991-t001:** *t*-test for the additive scales.

	Mean of Scale100 Boys	Mean of Scale100 Girls	*t*	α (2−code)
hostile sexism	60.7	48.9	17.03	0.000
benevolent sexism	66.9	67.6	0.13	0.713
ambivalent sexism	63.8	58.3	23.77	0.000
justification of domestic violence	32.5	25.1	34.96	0.000
justification of peer violence	37.2	29.1	11.35	0.001
justification of violence against minorities	40.4	39.7	0.16	0.690

**Table 2 ijerph-17-04991-t002:** R squares of latent variables—the mixed two step approach (boys and girls).

					95% CI	
	LV	R Square	Bootstrap	S.E. (Bootstrap)	LL	UL
Boys	ambivalent sexism	0.98	0.98	0.01	0.95	0.99
	justification of domestic violence	0.54	0.61	0.05	0.46	0.71
	justification of peer violence	0.57	0.62	0.05	0.51	0.75
	justification of violence against minorities	0.52	0.56	0.05	0.43	0.65
Girls	ambivalent sexism	0.97	0.97	0.01	0.95	0.98
	justification of domestic violence	0.69	0.71	0.05	0.60	0.80
	justification of peer violence	0.46	0.51	0.05	0.41	0.60
	justification of violence against minorities	0.61	0.64	0.04	0.53	0.72

S.E. = Standard Error; CI = Confidence Interval; LL = Lower Limit; UL = Upper Limit.

**Table 3 ijerph-17-04991-t003:** Path coefficients—the mixed two step approach (boys and girls).

					95% CI	
	LV	Path Coefficient	Bootstrap	S.E. Bootstrap	LL	UL
Boys	hostile sexism	0.85	0.18	0.07	0.01	0.32
	benevolent sexism	0.20	0.86	0.06	0.77	0.99
	justification of domestic violence	0.74	0.78	0.03	0.68	0.84
	justification of peer violence	0.75	0.79	0.03	0.72	0.86
	justification of violence against minorities	0.72	0.75	0.03	0.66	0.81
Girls	hostile sexism	0.57	0.51	0.01	0.49	0.53
	benevolent sexism	0.51	0.56	0.01	0.54	0.60
	justification of domestic violence	0.83	0.84	0.03	0.77	0.89
	justification of peer violence	0.68	0.72	0.03	0.64	0.77
	justification of violence against minorities	0.78	0.80	0.02	0.73	0.85

**Table 4 ijerph-17-04991-t004:** Girls against boys: differences in path coefficients.

Path Coefficient	Difference	*t*	*p*-Value
hostile sexism->ambivalent sexism	0.28	4.94	0.000
benevolent sexism->ambivalent sexism	0.31	4.26	0.000
ambivalent sexism->justification of domestic violence	0.09	2.01	0.045
ambivalent sexism->justification of peer violence	0.08	1.67	0.096
ambivalent sexism->justification of violence against minorities	0.06	1.51	0.132

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
