# Peer review of "The Use of Partial Least Squares–Path Modelling to Understand the Impact of Ambivalent Sexism on Violence-Justification among Adolescents"

_ijerph, 2020, doi:10.3390/ijerph17144991_

Round 1
Reviewer 1 Report
I appreciate the authors' extensive revisions which have greatly improved the paper. There are some minor changes that I recommend at this point:
Lines 162-164: Before giving the questionnaire to the students, we met their teachers and parents who provided the necessary consent for the students’ participation in the study, in accordance with the Italian Psychological Association ethical guidelines.
Please provide more details of what compliance with the Italian Psychological Association
ethical guidelines entail. Did you also obtain the informed consent of the research participants themselves (rather than just from the teachers and parents)?
I see that the questionnaires were self-administered after the students were reached between classes. What does that mean? Were they administered in a place where students had privacy? Or did they fill them out at their convenience? Did they take them home? Was it clear that they did not have to answer the questions or could skip questions? Was it clear that there was no coercion involved? Perhaps you could mention the 10% (or 40) who had more than 50% of data missing (that you excluded) as an indication that students felt free to skip questions that they did not want to answer.
Grammatical issues and the need for clarification are detailed below with the use of [brackets] to identify errors and recommended edits:
The Ambivalent Sexism Inventory (ASI) for adolescents, has been used to 23 evaluate the two dimensions of ambivalent sexism, i.e. hostile sexism (HS) and benevolent sexism 24 (BS). Moreover, the Questionnaire [don’t capitalize] on attitudes towards diversity and violence (CADV) was 25 administered to assess participants [missing apostrophe] attitudes towards violence.
Line 90: Recent systematic reviews highlight the topic showing that adolescents having more sexist attitude [should be attitudes] show a more positive orientation towards intimate partner violence,
The word “positive” is confusing in this context. Do you mean “reveal greater acceptance of intimate partner violence”? The same applies for the use of the word “positive” on line 102 (where there is another error):
Line 102: Carrascosa et al. [15] analysed the differences between boys and girls in their violent behaviour with respect [add word “to” here] their peers. In their study [add comma here] boys revealed higher positive attitude [cut “positive attitude towards” and say: “greater acceptance of” either rules transgression, either direct as well as indirect violence
Line 105, 122, and 123: researches [should be “research”] Used incorrectly in 3 places.
Line 105+: Other researches on adolescent behaviour showed that prejudices affect perceptions of violence and shape its definition [16, 17]. While some stereotypical beliefs about men and women’s roles, diversity and minorities, may justify violent attitudes [18, 19, 20]. [This should be one sentence. The second part starting with the word “While” is a sentence fragment].
Line 113: boys are more incline [inclined]
Lines 115-6: Recently in the Anglo-American literature, dating violence by the use of digital technologies and social media [CUT: obtained an increasing attention]. Instead say: [has received increased attention [or increased scrutiny]]
Line 119: This is confusing:
The well-known ideology [What well-known ideology??], which promotes asymmetrical relationships of power between men and women, often at the basis of the emergence of violent behaviours [do you mean: often is the basis or the driver of violent behaviours?]
Line 157: For the research here presented, [presented here]
Line 161: the word “housewife” is considered offensive and outdated by many. Say: had a stay-at-home mother or “had a mother who did not work outside the home”
Lines 198-9: These results are also better [stronger or more robust] than the ones provided by the original validation study [the word “better” implies a value judgment]
On page 9: You offer this comment: “Between August and December 1938 Italy adopted a series of legislative provisions that deprived Italian Jews of their civil rights and came to be known as the “Racial Laws”. The decrees of expulsion from school, linked to these laws, caused a deep wound in the life and identity of Italian Jewish children. This terrible page, in our history, identifies an open wound that we attempt to heal by making all students - of all grades and levels - study it. Therefore, the authors, do not believe it needed additional explanations either in the questionnaire or during its administration.”
I can now understand why this explanation was not required for your research participants. You do, however, need some explanation in your paper for your readers. It could even be in a short footnote. But there needs to be some explanation mentioned somewhere in the paper, even if only briefly.
Lines 284-5: Italian text requires translation (just an oversight, I’m sure)
Line 407: Ambivalent [don’t capitalize]
Line 413: do not only concern [apply to] adults.
Line 429: subtile. [subtle]
Line 435: Nevertheless, in our case the most alarming result concern [involves] girls.
Line 438: thereby accepting and even facilitating [CUT: in putting into action]. Just say: “and even facilitating them.”
Line 459: of outmost [utmost] importance
Line 466: Limitations and Future Developments
This limitation needs to be re-worded and doesn’t give the reader enough information.
I would cut this: [The main limitation of this study poses a threat to its external validity of the results and consists in the fact that the sample size does not allow the results to be generalized.]
And say instead: “This study sample has limited generalizability as research subjects were limited to be specific geographic region . . . . ”
In other words, what specific qualities of your sample matter and limit the generalizability? How might your sample be different from respondents in other parts of Italy or other countries? Just telling us it has limited generalizability does not provide any insight into specific aspects of your sample that could have influenced your results.
Line 474: Next hypothesis It should be: The next hypothesis . . .
Line 482: Don’t have a one-sentence paragraph. Simply include it with the previous paragraph.
Reviewer 2 Report
Thank you for taking into account the remarks and suggestions.
Author Response
Thank you very much for your valuable contribution and support.
Best regards,
The Authors.
Reviewer 3 Report
Thank you for the opportunity to review this paper. This is very important work. I have provided some insights into how to strengthen the argument. I also thank the other reviewers for providing excellent feedback and commend the authors for incorporating that feedback. This work is an important contribution. That said, the results section is a bit confusing and as such undermines the argument.
Abstract:
What is meant by sexism needs to be clearly defined in the abstract
What is the “gender relationship paradox”? and how might it be resolved on p. 2?
P2 line 68: needs rephrasing, does not follow English language guidelines, “this entails to see women as threaten..”
Suggest spelling out BS and HS, otherwise it gets very confusing.
P3 line 108: What is this paragraph staring with “In Spain…” trying to do? Is it trying to situate the Italian context within a larger global one? This paragraph needs some re-organizing to make clear what the point of it is. Right now it does not have focus and as such undermines the argument.
Moreover, there is one sentence with 5 citations, “Recently in the Anglo-American literature…” but what did these studies uncover? How do these studies specifically relate to the current one (i.e., is this paper testing hypotheses derived from these studies, is this paper replicating similar studies done elsewhere for the Italian context? What do “digital technologies and social media” have to do with dating violence, studying dating violence, with this paper?
- 3 line 122: the use of researches here needs work to follow English language guidelines.
P.3 line 125-135: consider organizing this instead of – with numbers i.e., aim 1, aim 2, hypotheses 1, 2.
What does a different composition of Ambivalent sexism mean? What would it look like? More information is needed here as it is not self-evident.
What does the CADV have to do with this study? Is it the same questionnaire being used? Please say so. Has been validated? What are the validation measures?
Paragraph starting line 141 needs more relevant information on why these latent variables are good measures of the concepts being measured and how they related to the observed variables from the different inventories being used.
How does the “ultimate goal” at the top of p.4 147 possible given this research? How does studying benevolent sexism and hostile sexism create specific actions to combat and prevent violence among adolescents? More is needed to link these objectives in the paper, if this is the main objective.
Is there actually a measure of how many used gender-based violence? Here (line159) it says to justify gender-based violence; is that for them? Hypothetical situation? In the media?
Line 163 p. 4, says within IPA ethical guidelines, was their IRB? Why or why not (this will be helpful information for an international audience not familiar with the Italian context.
Consider moving the validation of the survey line 198-99 on p. 5 earlier to discussion of the inventory (see earlier comment)
Line 222: suggest spelling out all the acronyms for SEM-PLS and SEM-CV, PLS-PM, LV etc. the first time they are used. It is confusing as is.
I am a bit concerned about the degrees of freedom with less than 400 cases and 67 observed variables to make up the latent variables.
Line 263, justifications for use of the model echos added paragraph line 222. Some of this is redundant or could be better synthesized and articulated potentially as one section or together.
Line 284: there is a sentence in Italian, not sure if it for the Italian version of this article.
What is being predicted? What is the response or dependent variable?
Author Response
Please see the attached file.

This manuscript is a resubmission of an earlier submission. The following is a list of the peer review reports and author responses from that submission.
Round 1
Reviewer 1 Report
I believe that this article will make an important contribution to the literature. I was glad to have the opportunity to review it and think that with some minor additions, it will be a strong paper.
Here are my suggestions, listed in order as I read the paper:
1. Change the 2nd part of title and mention instead something about benevolent sexism as eliciting equal resentment, perhaps specifically include the words ambivalent sexism. The current 2nd part of the title is off-putting, and I wouldn't have read the paper as a reader (versus as a reviewer) with that title and would be much more interested in reading the paper if I knew a bit more about the topic other than the general terms "Sexism and Violence" that are in the first part of the title.
2. Change wording on lines 20-22:
I SUGGEST THIS WORDING INSTEAD:
A partial least square - second-order path model - reveals that girls resent benevolent sexism as much as hostile sexism, while among boys, there was more a disparity in how they viewed the two types of sexism.
3. Line 33: "comprises "should be "comprised of "and don’t have a one-sentence paragraph (as I see also on line 44)
4. Line 97: Starting from this research
Do you mean: “In this research” . . . ??
5. Line 108: “is" should be “was”
6. Lines 127-8: Before giving the questionnaire to the students, we met the students’ teachers and their parents who provided the necessary consent for the students’ participation in the study.
Was this research reviewed and approved by an ethics board/IRB?
7. "The questionnaires with more than 50% of data missing were eliminated; finally, 360 questionnaires were collected and used for the analysis."
So we need to know that the final sample size of 360 was out of how many? It’s not clear what proportion had to be thrown out due to missing data.
8. Line 137: You say, “The questionnaires were self-administered” BUT then you say on line 141 that, “This form of administration required questions and answer alternatives to be read, for everyone, by an interviewer.”
If the interviewer had to read questions and answer options, we need comprehensive details about who the interviewers were (as much demographic info as possible), where the interviews took place, how many agreed to participate, how long the interviews took, etc.
We also need to know whether some respondents were interviewed by males versus females, etc. I.e., for such sensitive data, was there any attempt to match demographic variables of interviewer and interviewee.
9. Was there any incentive for students to participate?
10. On p. 6, variable 34: Asking for help is a sign of strength.
Was that reverse coded? Otherwise it doesn’t make sense.
11. Variables 58 and 59 deal with attitudes about Jewish people. But there is no discussion of anti-Semitism in the paper. Why were attitudes about Jewish people important to study in Italy? One item refers to the “expulsion” of Jews. That requires an explanation. Are we sure that respondents understood that history?
Perhaps issues surrounding these questions about Jewish people could be incorporated into the limitations, a section is missing.
12. The authors say, “The present study has some limitations” yet none of those limitations are specifically articulated.
Good luck with the revisions.
Reviewer 2 Report
Sexism and violence among adolescents: The use of Partial Least Squares-Path Modeling to understand the gender differences
Abstract: The presentation isn’t complete for a scientific article. For example, there is no information about instruments.
Introduction
Only two papers are cited about sexism and violence of adolescents or dating relationships. Authors may wish to include several references about sexism of adolescents or young people:
Ferragut, M. (2014). Analysis of adolescent profiles by gender: Strength, attitudes toward violence and sexism. The Spanish Journal of Psychology, 17, E59. doi:10.1017/sjp.2014.60
Ibabe, I., Arnoso, A., & Elgorriaga, E. (2016). Ambivalent Sexism Inventory: Adaptation to Basque population and sexism as a risk factor of dating violence. The Spanish Journal of Psychology, 19, E78. doi:10.1017/sjp.2016.80
Lopez, V., Chesney-Lind, M., & Foley, J. (2012). Relationship power, control, and dating violence among Latina girls. Violence Against Women, 18(6), 681–690. https://doi.org/10.1177/1077801212454112
Ramiro-Sánchez, T., Ramiro, M. T., Bermúdez, M. P., & Buela-Casal, G. (2018). Sexism in adolescent relationships: A systematic review. Psychosocial Intervention, 27(3), 123–132. https://doi.org/10.5093/pi2018a19
Lines 97-104: Starting from this research, we have studied sexist attitudes among adolescents. First, we have related sexist attitudes to the justification of violent attitudes against women, children, peers and minorities. Using the inventory of ambivalent sexism …
I think that there is any information in this paragraph to integrate in Method section.
Objectives
I think the motivations for this study need to be made clearer. In particular, stating the objectives more precisely and highlighting the novelty of the study. Moreover, it would be interesting to put forward some hypotheses.
Lines 105-109: It is necessary to rewrite the objective of the study as well as to include its relevance: The goal of this analysis has been to analyse the association between sexist attitudes and the justification of violent attitudes among adolescents, to understand how they experience being in a couple and acting the differences between genders, as well as to investigate if and to what extent they contain forms of aggression and violence
For example: The goal of the present study …
Method
Lines 121-123: Originally, it was believed that the phenomenon of gender violence only concerns adults. Today numerous empirical studies have shown that it is also perpetrated within relationships described as relating to protagonist teenagers.
This information is not appropriate for this section because Method should describe how the study was conducted. It would be recommendable to move Introduction.
Where is the description of sample?
Pages 3-4: 2.1. Measurement instruments
This section doesn’t follow APA recommendations: description of instruments and their psychometric properties.
2.2 Methodology or Data Analysis? Authors may wish to mention in this section the advantages of SEM over PLS-SEM. All tables and figure are not essential.
Pages 6-8: Table could be deleted or put as an Appendix. Figure 2 and 3 are very similar. Ambivalent sexism had no associated manifest variables associated Is this correct? Page 12: the same influence (0.52) as 288 hostile sexism (0.57).
The Results and Discussion section is difficult to follow. I think that a more in-depth discussion of Figs. 4 and 5 would be helpful. In this section there isn’t discussion, because the results found are not discussed in relation with previous studies.
Page 12: the same influence (0.52) as 288 hostile sexism (0.57). The term “influence” is not appropriate when the design is not experimental, because it isn’t possible to study the causality.
Conclusions This result suggests some implications for prevention interventions, which should aim at changing attitudes toward gender roles in adolescence. Thus, interventions should target the males’ sexist attitudes, but also those of girls who believe they must be submissive to boys and to develop an ecological intervention The conclusions are too general and nothing original.
Reviewer 3 Report
The proposed manuscript investigates the relationship between sexist attitudes and the justification of violent attitudes between adolescents. Their experience of being in couples and the differences between genders are studied in relation to possible occurrence of aggression and violence. A representative data set is analyzed statistically by using Partial Least Squares-Path Modeling approach.
The presentation of the results is clear and comprehensive. The results are valuable and worthy of being published. The obtained results can be used for improving social relationships and communications in the contemporary society.
Minor revisions are suggested to improve the quality of the exposition.
- p.1, l. 26: It should be “wrote:” instead of “wrote”;
- p.2, l. 52: It should be “wrote:” instead of “wrote”;
- p.3, l. 120: The ending dot of the sentence should be after the citation.
- p.4, l. before 169: the lower and upper bounds of the sum should be written under and above the sum sign;
- p.4, l. after 110 (the last sentence): It should be “the derivative with respect to time” instead of “the derivative of time”
- p.4, l. 179: It should be “32]” instead of “32)”
- p.4, l. 182-183: The subscripts of \sigma should be written properly: \sigma_1, etc.
- p.9, l. 233: It should be “MVs standardized” instead of “MVs a standardized”
- p.13, Table 8: The values for x_11, x_21, x_38 and x_54 are not written very precise